# A Dispersion Index for the Analysis of the Distribution of Activities in the Tunisian Coastal City of Nabeul

**Narjiss Bakhtyari** [1,2,3,*] **, Asma Rejeb Bouzgarrou** [1,2,3] **, Christophe Claramunt** [2] **and Hichem Rejeb** [3]

1    National School of Architecture and Urban Planning, University of Carthage, Carthage 1054, Tunisia; asma.rejebbouzgarrou@enau.ucar.tn
2    Naval Academy Research Institute, Lanvéoc-Poulmic, 29240 Brest, France; christophe.claramunt@ecole-navale.fr
3    Research Unit Horticulture, Paysage et Environnement, Higher Agronomic Institute of Chott-Mariem, Sousse University, Sousse 4002, Tunisia; rejeb.hichem@iresa.agrinet.tn
*    Correspondence: narjiss.bakhtyari@etudiant.univ-brest.fr

**Abstract:** This research investigates the duality of the spatial organization and urban activities generated by the coastalization processes of the city of Nabeul. The first part of the study analyzes the city's fragmentations of the urban landscape using a novel generic index of directional dispersion and a set of space syntax metrics. These structural and functional properties are studied by the concentration and/or dispersion of urban functions of the evolution and development patterns. Among the emerging features, we observe a dispersion of urban activities beyond a central radius all along the western periphery of the city, confirming the phenomenon of urban sprawl that many Tunisian cities are experiencing. These spaces generate urban fragmentations of "new polarity zones" under the influence of the coastline attraction. Finally, this study introduces a novel approach for identifying urban structural polarity and activities, as well as new perspectives for coastal land management and planning.

**Keywords:** coastal urbanization; urban dualities; space syntax; index of dispersion; Nabeul

## 1. Introduction

In recent decades, humankind has witnessed dramatic population shifts from rural to urban areas [1]. According to the 2011 Revision of World Urbanization Prospects [2], the global urban population increased from 0.75 billion (29.4% of the world's population) in 1950 to 3.63 billion (52.1%) in 2011 and is predicted to reach 6.25 billion (67.2%) in 2050. Coastal cities are significantly impacted by these urban trends and have been the object of substantial structural alterations and even self-reorganization processes. Mediterranean coastal locations are very appealing due to their year-round pleasant weather [3]. Therefore, the fast-growing development of seaside tourism has largely contributed to the image of the coastline and its urban dynamism [4]. Several works have studied urban sprawl phenomena and have shown that coastal urbanization has a significant and direct influence on the emergence of socio-spatial distancing [5]. The southern Mediterranean coastlines are currently subject to significant changes with a range of pressures related to population growth [6], city development and entertainment zones that characterize these areas [7,8]. Coastal areas, through their multifunctional attractive tropisms [9], reflect seaside territorialization that is poorly or not at all integrated with urban centrality [10]. Such development may result in urban disparities and lifestyle changes in these privileged spaces. New practices and uses are frequently observed, resulting in spatial heterogeneities, causing a fragmentation of urban configurations [11,12]. Furthermore, many researchers have attempted to better comprehend coastal areas in different contexts, emphasizing the correlation between littoralization and "dispersed urbanization" [13]. Finally, it has

been observed that the development of coastal cities has been related to a spatial-temporal fragmentation [14] characterized by a central and littoral fracture.

Historically, the urbanization of Mediterranean coastal cities has always progressed in a linear direction along the coast [14]. However, when coastal areas reach their land saturation, this generally generates the displacement of growth expansion to inland territories which become attractive again for specific urban purposes [15]. Other researchers have shown that inland growth tends to expand preferably along the roads [16]. Moreover, urban networks not only act as material support for urbanization but also as an element of urban space structuration. In fact, the urbanization of coastal cities is supported by new settlements near major roads, which regularly shape and structure the city boundaries. This phenomenon is sustained by forces of dispersion that lead to the emergence of new polarities with unequal disparities, more or less distant from the city center, but reasonably well connected [17]. Based on the hypothesis that each socio-spatial configuration generates specific properties and relationships [18], the basic idea of this study is to introduce an appropriate model that reveals potential causal dependencies between spatial configuration and urban functions that may be used to support the emergence of a center-coastal disparity.

### 1.1. Configuration of the Urban Network in Coastal Cities

A thorough modeling of urban growth processes is a prerequisite to a sound understanding of urban dynamics, as well as better management and planning tasks [19,20]. Several models have been developed to describe urban structural properties, including the concentric zone [21], sector based [22], multi-nuclei [23] and central place approaches [24]. Recent urban sprawl studies simulated land-use changes using cellular automata, complex systems theory and agent-based systems and characterized them at a fine spatio-temporal level [25,26]. However, most of these works concentrate on urban structural properties [1–27] rather than socio-economic patterns and human behaviors [19]. Therefore, several lines of research have explored a closer connection between spatial configuration, land use and urban economics [28]. Our research is grounded on Space Syntax that offers a syntactic language that quantitatively describes urban spatial structure and considers human spatial behavior as a sort of natural movement [27–29]. Space syntax offers valuable mathematical and computational resources for studying the spatial structure of a specific urban environment in relation with functional properties from transportation, land use and human behavior [30]. Many space syntax studies based on the urban network modeled as an axial map, which is often the main structural layout considered, have demonstrated that activity locations are drawn to the most spatially integrated streets at different scales [18,31–35]. Some of these studies also revealed new forms of spatial organization, namely concentrations, polarities and centralities [18].

Space syntax practically evaluates the relationship between streets and their surrounding urban functions using a series of metric measurements to quantify various syntactic measures and values associated with axial structures such as connectivity, integration and choice [36].

Despite their relevance, and to the best of our knowledge, these studies examine the road network rather than the street directions and associated metrics, which are important factors when studying urban disparities. This leads us to consider the fact that roads embed both distances and directions and to introduce a new structural index of directional distribution of activities inspired by statistical metrics and applied to Nabeul as a case study. These variables, combined with common space syntax indices, may give a "reading key" for a better understanding of these structures' orientation. To practically explore the relationship between the urban network and the distribution of urban functions, we model two structural and functional characteristics: (i) the concentration of urban functions and (ii) their dispersion throughout the study area. Several metrics are for instance available in economic geography to qualify some measures of spatial concentrations of economic activities [37] but different distributions may result in similar interpretations, and they are also sensitive to isolated values, and so far, they have hardly been applied to the urban domain.

The added value of this study is that it provides a generic Directional Dispersion Index of the spatial distribution of activities in an urban environment, which describes the disparity process by considering the directional attribute of the urban network. This structural approach associated with this analysis tends to give us additional clues on spatial practices in relation with the degree of connectivity and continuity of specific city areas [38]. This study aims to highlight the findings obtained by comparing the measures of urban activity distribution and structural properties and then reveal a sort of spatial opposition in the organization and frequentation of the central and coastal areas. These facts should provide a better understanding and explanation of current urban findings by emphasizing the primary local and global characteristics of the road structure in the city of Nabeul for a better territorial reorganization [39].

### 1.2. Research Case: The City of Nabeul

Nabeul, like many other Tunisian coastal cities, suffers severe issues as a result of a fragmented and dysfunctional urban structure that impacts its management. Several studies related to the city of Nabeul explored landscape transformations and found a distinct spatial organization associated with littoralization [7,40,41]. Nabeul is one of the most important cities on Tunisia's north-eastern coast, located south of the Cap Bon peninsula and 67 km southeast of the capital Tunis. It is bordered in the south by the Mediterranean Sea, on the east by the town of Dar Chaabane El Fihri and on the west by Hammamet city (Figure 1).

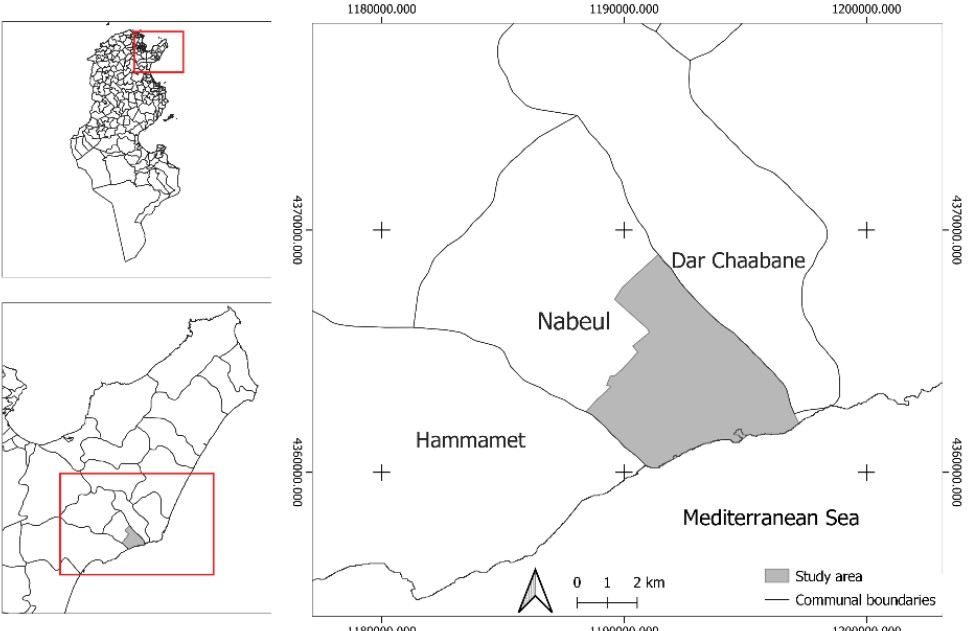

**Figure 1.** Situation map of Nabeul city.

Our study area is the capital of the governorate of Nabeul, with 73,128 inhabitants in 2014, 96% of whom live in communal areas. Nabeul has a greater growth than the national average (1.28% compared to 1.25% for the period from 2004 to 2014) [42]. Since 1975, the city of Nabeul has evolved and transformed in response to tourism activities, which contribute to the differentiation of its urban structure between the historical core (medina) and the urbanization of its coastline [43]. This touristic dynamic stimulated internal migration (which increased from 1.9% in 2004 to 4.6% in 2014) and had an impact on the urban area, causing the city to sprawl (Figure 2).

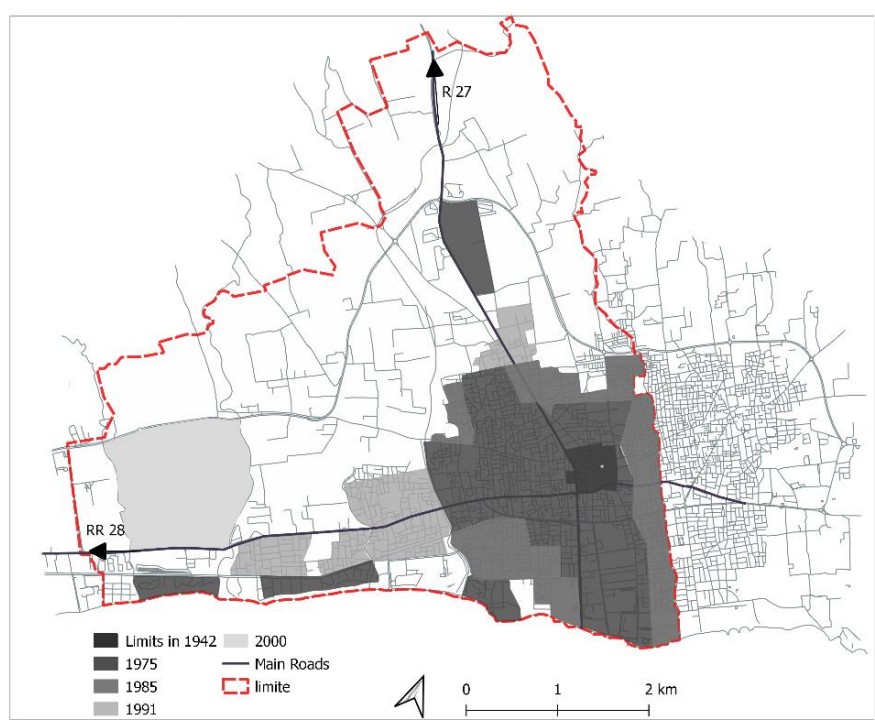

**Figure 2.** Urban development of Nabeul city.

This expanded character of the urbanization was supported by new localizations near the main touristic roads (the regional road, RR 28), which progressively and continually displaced the city's boundaries in favor of a coastal conurbation (Al Mrazga) (Figure 3).

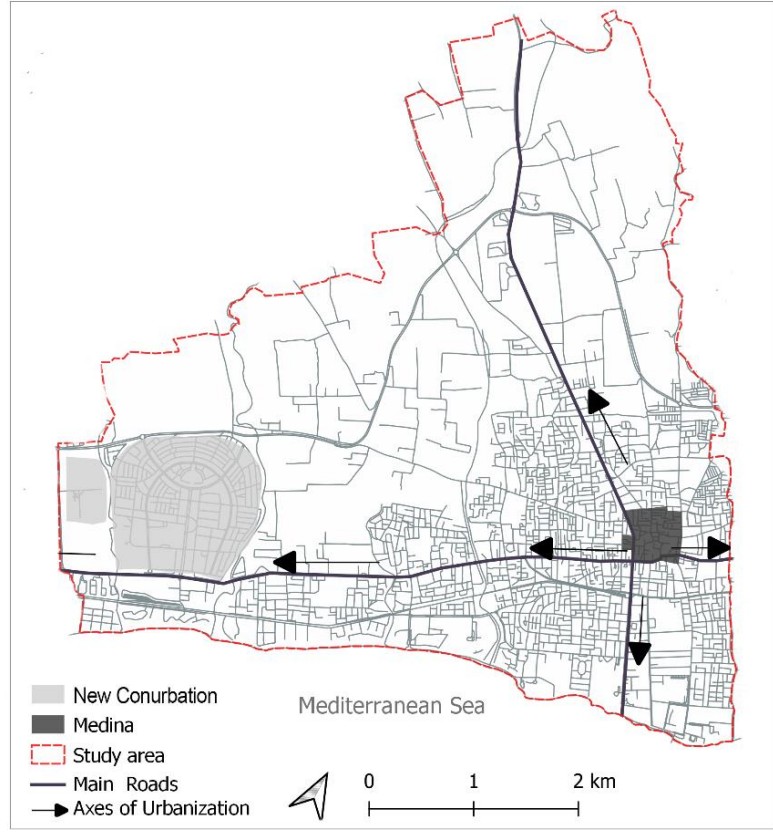

**Figure 3.** Map representing the main urbanization axis on the western edge.

The remainder of this study is structured as follows. Section 2 discusses the main principles underlying our methodology, including a review of current research as well as the basic principles of dispersal indexes and space syntax theory. Therefore, we introduce a dispersion index based on a discretization of distance to the center of commercial and service activities, in conjunction with an axial analysis of the urban road structure, which provides complementary measures of integration and choice. Section 3 describes the key results and findings of our study as applied to the city of Nabeul. Section 4 discusses the findings and we finally conclude the study in Section 5 with some final insights and perspectives.

## 2. Materials and Methods

With an emphasis on the spatial effect of littoralization, our approach aims to comprehend in novel ways the urban organization of a coastal city, which implies an unequal spatial distribution of urban activities. In other words, we believe that urban sprawl is related to the coastline's attractive forces and urbanization, which are expressed by dispersion of activities from the center to the coastline. In this context, two methods are applied when carrying out this spatial duality. The first method introduces a concentration-dispersion using the linear location of commercial activities on road networks. The statistical figures are derived by a Directional Dispersion Index at a recent date (2018).

The second method explores configurational attributes of the road network using space syntax measures based on the assumption that a road network can be represented as a weighted planar graph [1]. The axial map generates an analysis of the spatial accessibility of the road network. Structural metrics such as integration and choice are applied at the local and global levels within a diachronic evolution (1975–2018). The years from 1975 to 2018 can help us understand the historic development process and thus capture the spatial structure's evolution. The year 1975 refers to a period of urbanization marked by tourist and industrial expansion, as a result of government economic policies conducted in the 1970s [7,44,45]. The year 2018 is then chosen as a representative recent date on which Nabeul has experienced extremely violent storms and torrential rainfall that have challenged the urban planning of the city. All geographical data are based on aerial photographs from 1975, as well as prior research [7] and recent Google Earth images. Lastly, the confrontation of statistical indexes with structural analysis investigates the relationship between the distribution of commercial activities and the emergence of new forms of spatial organization induced by coastal attractiveness.

### 2.1. Construction of a Combined Urban Distribution Model

Several works in different contexts introduced various measures of concentrations or dispersions. Economic geography has introduced some measures of geographical concentration of economic activities such as the locational Gini and Herfindahl indices [46]. These indices are useful for understanding the intra-sectoral distributions of economic structures, but they do not consider spatial information. In fact, they are insensitive to the spatial dimension [47], despite the fact that such distributions might be evaluated using quantitative measures. Location-based measures can estimate the central tendency of a given cluster (e.g., mean and median). For instance, in a related work, in order to evaluate the organization of an urban space around different centers, the GEODE research unit developed an approach to identify the geographic/concentration center of a set of entities by applying a Mean Center and Standard Distance measures that take into account distances to the center in relation to the average [48]. This led to concentration and dispersion indices, which denote the spread of some considered values in a given distribution. The greater the value of the dispersion metric, the more spread out the distribution is. Based on these principles, but also taking into account additional directional properties, the model developed in this study introduces and combines directional dispersion indices with structural properties of the urban network, such as integration and choice, which are two commonly used versions of centrality metrics applied in spatial syntax to express the link between the urban layout and its functional model (e.g., human movement).

The approach is applied to the urban network and activities of the city of Nabeul. Both structural and functional metrics are combined to highlight the links between the dispersion and concentration of activities in the city of Nabeul, and thus to derive the logic of their urban distribution (Figure 4). Indeed, the spatial distributions of a variety of activities is computationally evaluated in relation to the city of Nabeul's core. This should lead to a better understanding of the mechanisms that denote the reconfiguration or spatial fragmentation of urbanization processes. Accordingly, this led to the derivation of a series of specific measures to capture this duality. The main theoretical contribution of this study is a generic index of directional dispersion inspired by both the standard deviation and the interquartile range which serve to characterize the spread of values present in a distribution but also by the directional distribution which consists in summarizing the spatial characteristics of the activities in relation to the city center and according to dispersion and directional patterns. Finally, combining the statistical index and structural analysis within a single approach, as depicted by Figure 4, allows for an integration of complementary geographic and socio-economic data with spatial syntax figures [49]. This enables us to investigate the relationship between commercial activity distribution and the emergence of new forms of spatial organization caused by coastal attractiveness.

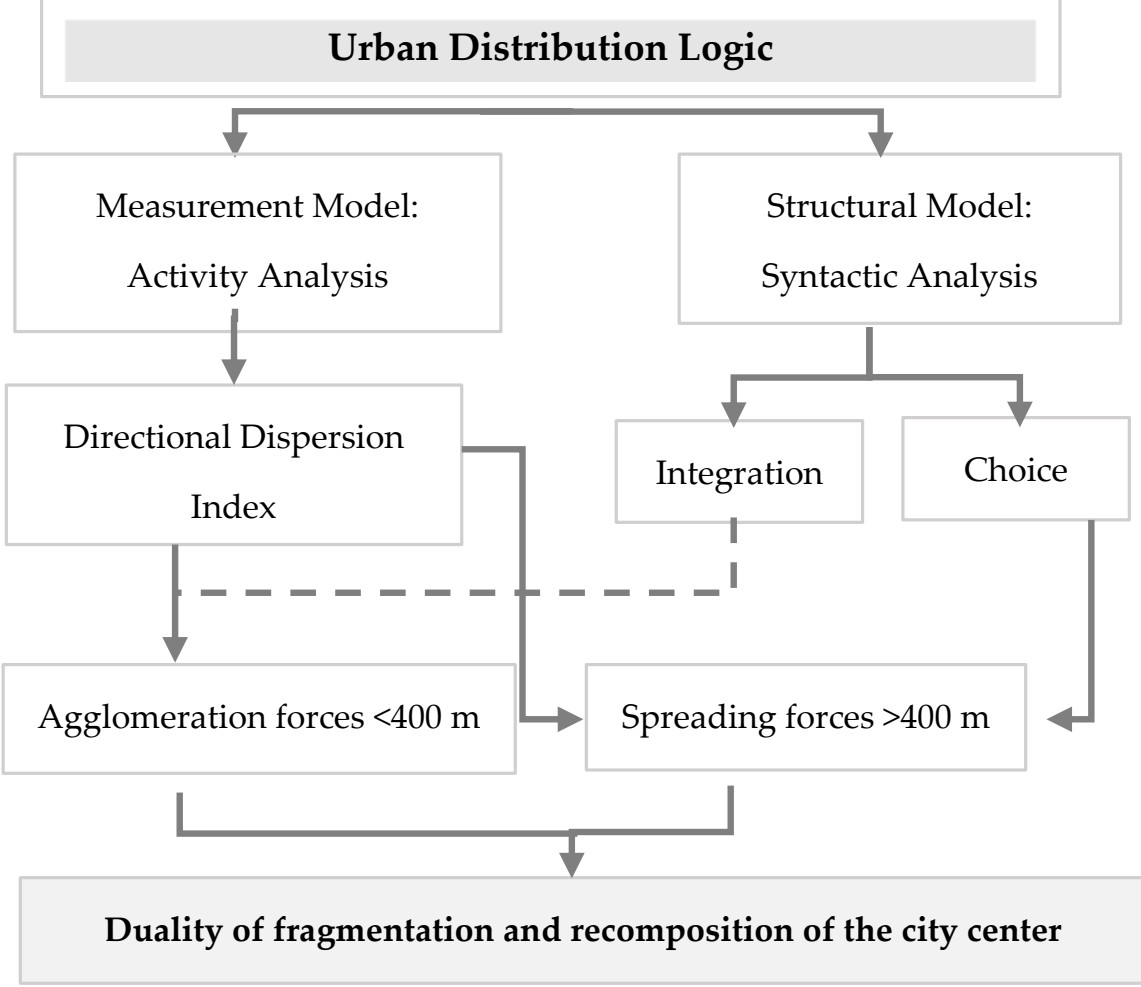

**Figure 4.** Conceptual approach of a fragmentation/recomposition dualism of new differentiated urban forms.

### 2.2. Searching for a Directional Dispersion Index

Our objective is to identify an index that characterizes the city's activity dispersion. Without loss of generality, let us assume that the concentration or dispersion of a structure or a set is determined by the number of units (N) that constitute it, as well as their distribution relative to a central value. This center is identified as a fictive point that is part of the ancient fabric and delimited by the intersection of the structuring roads; it is thus considered as the point with central coordinates and all urban activities are geo-located in reference to it. To calculate this index, we suggest the following method: it essentially consists in representing a number of given activities (N) by a subset of locations (xi) along a segment of axis (j). A distance (di) is assigned to each location, and the dispersion of this subset around the average distance (dmi) is then determined. Thus, an axial dispersion index of activities ID(j) is given by the means and standard deviation of the activity's distribution. Thus, it can be defined as follows:

$$ID(j) = \sqrt{\frac{\sum_{i=1}^{N}(di - dmi)^2}{N_j}} \tag{1}$$

with

$$dmi = \frac{\sum_{i=1}^{N} di}{N_j}$$

where di is the distance to the center (X,Y) of activity (xi) on axis (j) and (i = 1, 2,..., N), dmi denote the average distance to the considered center.

The dispersion of a number N of activities (xi) on one of the city's axes can then be evaluated. This can be schematized as follows:

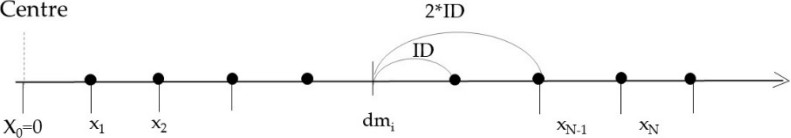

with di = $x_i - x_0$.

We use the same guidelines (1) and consider the main axes in the road network to which we assign sub-directions (cardinal directions) (Figure 5). This gives a Directional Dispersion Index for the four axes (east), (west), (north) and (south) and for the four orientations (2). These axes do not have the same length; therefore, we normalize these 4 indexes by a weighting factor

$$\sqrt{\frac{1}{D_{ij}}}$$

which for the (j) axis is:

$$IDD(j)_{norm} = \sqrt{\frac{\sum_{i=1}^{N}(di - dmi)^2}{N_j * D_{ij}}} \tag{2}$$

where $D_{ij}$ denotes the length of the axis.

After calculating the IDD(j)norm of every axis, the rate index can be aggregated and normalized. The result is expressed as a raw number. To express it as a percentage, the formulae is:

$$AggIDD(j) = \frac{IDD(j)norm}{\sum_{j}^{4} IDD(j)norm} \times 100 \tag{3}$$

AggIDD(j) is essential to explain the ratio between the total value that includes all axes and the partial value of each axis. Let us assume a distribution of N activities divided by the number of functional sectors. The first step is to quantify the existing facilities in the city of Nabeul, classifying them according to functional categories (equipment, services, commercial), as well as their distance from the center, in order to derive the central character of each type of polarizing function. The objective is to geolocate the urban

activities (Figure 6) and determine distributions using circular subdivisions of respective radii of 400, 800, 1200 m and so on (Figure 7).

In the context of a monocentric organization, an appropriate threshold distance should be specified for the core area. In this case, and without loss of generality, let us consider the central area of Nabeul as a circular area with an estimated radius of 400 m (Figure 5), which represents 50 ha. We observe that commercial facilities are prevalent throughout the city, even if the number of shops varies. Below the threshold of 400 m, the center concentrates a large number of commercial and service activities, and above this threshold, there is a significant decline in the number of these activities and a gradual increase towards the city boundaries (Figure 7). The focus was on commercial and service activities as an essential key to the study of the city [50] and, above all, as a generator of interactions and urban mobility.

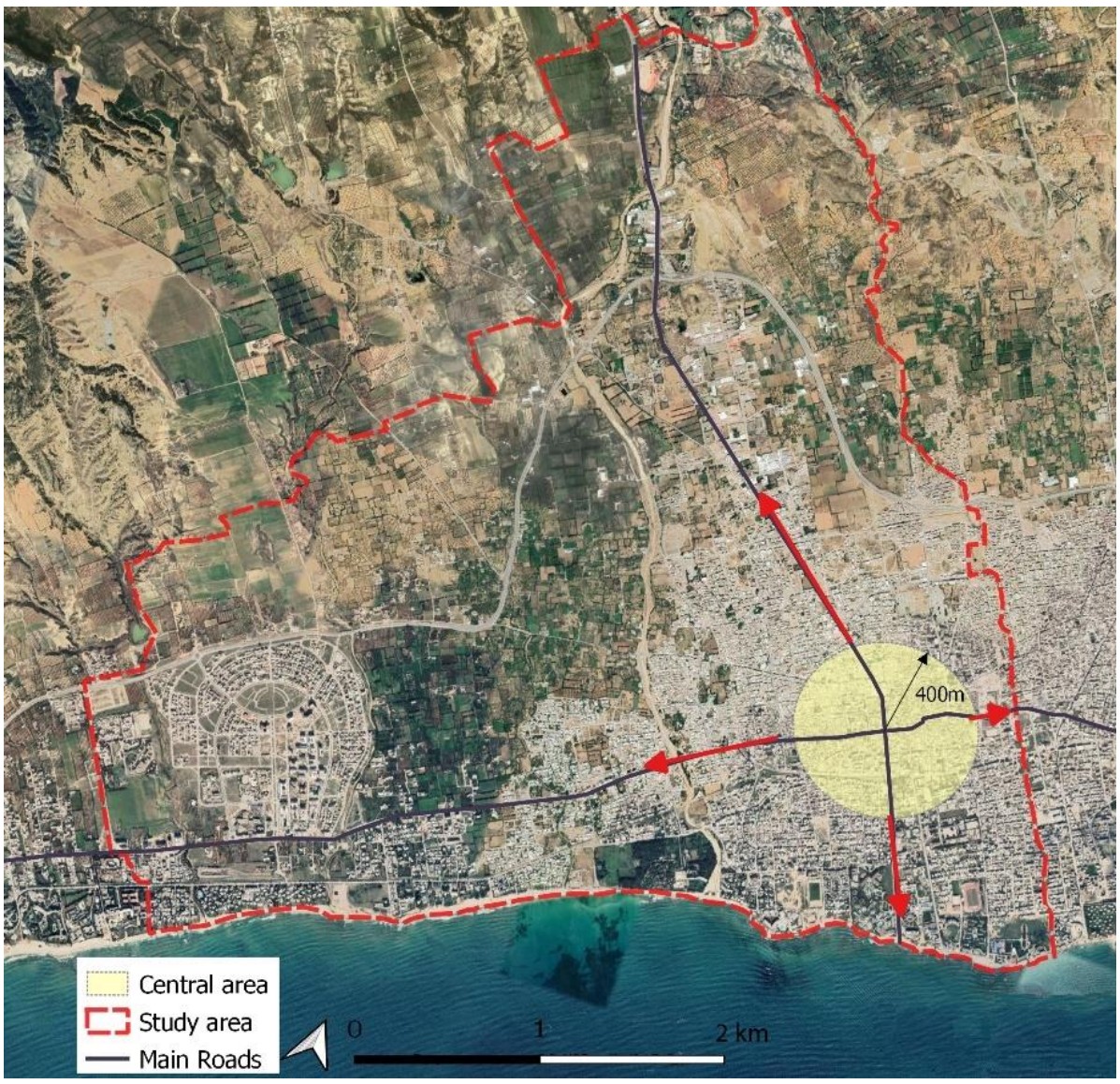

**Figure 5.** Study area and location of the center and the main axes of the road network.

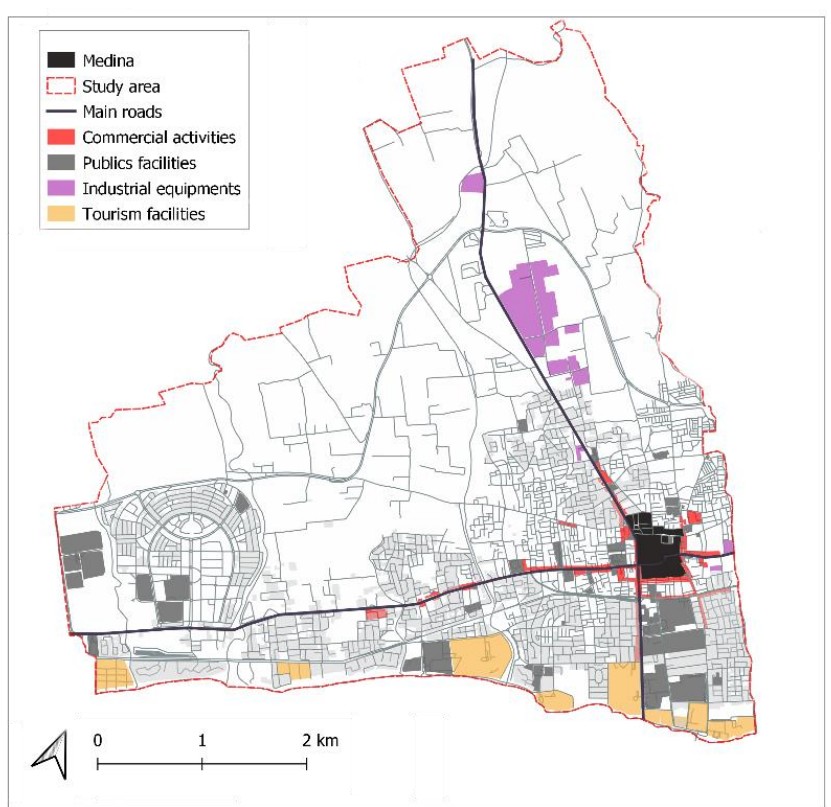

**Figure 6.** Location map of the different sectors of activity, Nabeul 2018.

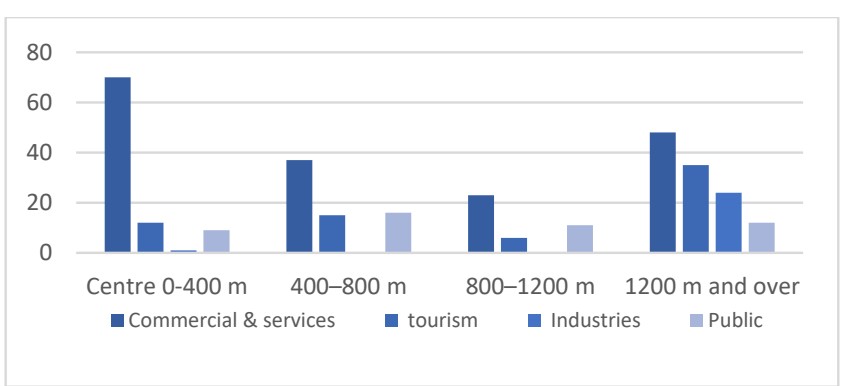

**Figure 7.** Distribution of the economic and public sectors according to distance from the center. Nabeul 2018.

Accordingly, we consider a central zone with a radius of 400 m (periphery beyond these 400 m) and a fictive central point. As an example, let us consider the representation of the index of the dispersion of commercial and service activities on the eastern axis. We consider a segment $[x_0, x_{n'}]$ and a subdivision of this segment into two parts $[x_0, x_n]$ and $[x_n, x_{n'}]$. There are N $(n + n')$ activities on the segment (axis), spread over the two parts (4) and (5).

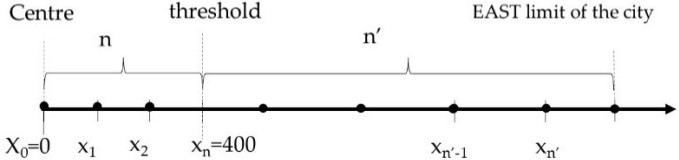

For the segment $[x_0, x_n]$:

$$\text{IDD(east)} = \sqrt{\frac{\sum_{i=1}^{n}(di - dmi)^2}{n * D_{ij}}} \tag{4}$$

with

$$x_i \leq x_n = {}_{400}, d_i = x_i - x_0, D_{ij} = x_n - x_0 \ (i = 1,2, \dots , n)$$

For the segment $[x_n, x_n']$:

$$\text{IDD(east)} = \sqrt{\frac{\sum_{i'=n+1}^{n'}(di' - dmi')^2}{n' * D_{i'j}}} \tag{5}$$

and

$$x_{i'} > x_n = {}_{400}, di' = x_{i'} - x_n, D_{i'j} = x_{n'} - x_n \ (i' = n + 1, n + 2, \dots , n')$$

It appears that the Directional Dispersion Index $({}_2)$ is sufficient in terms of what one could expect from an index of spatial distribution. The interest is dual: the directional dispersion index, while being generic, is configured according to the specific configuration of the structure of the city of Nabeul, yet this choice remains generalizable in other cases of study (i.e., four directional axes are considered as well as a central area of 400 m, but these are contextually defined for the city of Nabeul and then they can be valued according to each specific city's properties).

### 2.3. Syntactic Analysis

In order to understand the urban structure of the city of Nabeul as a whole, two maps at different dates were considered to depict the city's urban evolution. The road network was reproduced from the urban development plans of the year 1975 and 2000, then validated and completed using OpenStreetMap and Google Street View functions. Then, the axial map of the entire city was generated by the Depthmap$_2$ software. The syntactic analysis was performed on an axial map represented by a minimal set of axial lines used to shape the urban system's spatial structure. Three kinds of accessibility measurable indices were adopted: the measure of integration of an axial line represents the degree of integration or segregation of an axis in the network and evaluates the to-movement potential of a given line. A diachronic analysis approach was adopted, varying the dates between 1975 and 2018 on a global (Rn) and local scale (radii R$_3$). The measure of choice was used to identify the importance of the axis as a transit route for the network by representing the street segments that are highly chosen to be crossed as the shortest path.

The integration is based on the depth measure which is valued as the number of steps to reach the other axial lines for a given specific axial line (i.e., a step represents the un-weighted distance between two axial lines that intersect in the axial map [16]. The measure of integration of a given axial line (i) can be derived using the relative asymmetry as introduced by Hillier and Hanson [27], which compares how deep the system is from a particular point to how deep or shallow it theoretically could be. The formula is as follows:

$$\text{RA}_i = \frac{2(\text{MD}_i - 1)}{n - 2} \tag{6}$$

with RA: relative asymmetry = integration

MD: the average depth $\text{MD}_i = \frac{\sum_{j=1}^{n} d_{ij}}{n-1}$

n: the total number of the lines in axial map and $d_{ij}$ the depth of the ith axial line from the jth axial line.

Practically, a measure of integration can be applied locally or globally depending on the extent of the network and then nodes of the urban network layout can be considered, that is, the local integration limits the number of surrounding nodes considered [51]. The

local integration is usually calculated with 3 or 2 radii [52]. Once the syntactic properties are computed, a differentiation is performed to distinguish between the most integrated and the most segregated axial lines. Finally, the measure of choice, as introduced by Hillier et al. in a study entitled "Creating life: or does architecture determine anything?" [53], and overall, widely applied [30], identifies the shortest paths passing through one given node as a measure of connection of that given node to the remaining space [54]. This measure of choice appears to reflect the paths of people with specific knowledge of the urban spaces under study, including their residents. It is a measure of what researchers in spatial syntax refer to as through-movement potential. The measure of choice is given as follows:

$$C_B(P_i) = \sum_j \sum_k \frac{\sigma_{jk}(P_i)}{\sigma_{jk}} (j \neq k) \tag{7}$$

where $\sigma_{jk}(P_i)$ is the total number of shortest paths from nodes $P_j$ to $P_k$ and $\sigma_{jk}$ is the number of those paths through $P_i$.

These two measures represent two complementary forms of movement in an urban network by evaluating either the potential shortest movements to all other locations in the graph (i.e., integration) or the capacity of a given location to lie in a path of all shortest paths between all locations of the graph (i.e., choice) [18].

## 3. Experimental Results

### 3.1. Distribution of Commercial and Service Activities in the Study Area

The representation of the urban network's key axes reveals an apparent concentration of activities in the central area, as well as a linear distribution denoting a spread of urban activities along the main road axes (Figure 8). The application of the Directional Dispersion Index allows for the measurement of distribution of activities and also tends to reveal that commercial and service activities have diverse distribution patterns on the different axes that characterize the structure of the urban network.

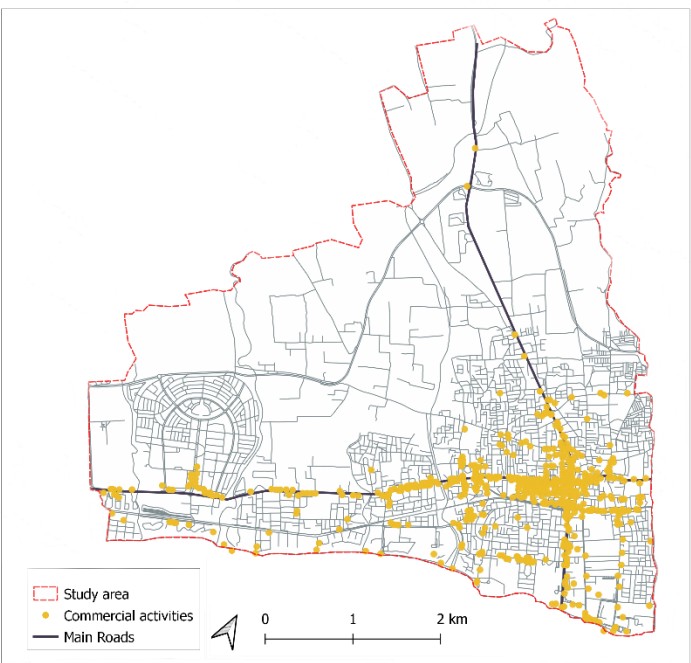

**Figure 8.** Mapping distributions of commercial and services facilities in Nabeul city.

### 3.2. Urbanization Diffusion along West Axis

Figure 9 shows an important contrast between the road axes with a very high value of the Directional Dispersion Rate on the western axis (47%).

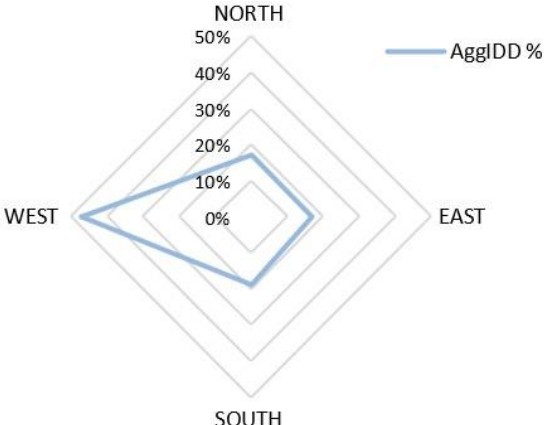

**Figure 9.** Directional Dispersion Index on the north-south, east-west axes.

It appears from these figures that the development of activities along the west axis is sustained with a dispersed character with a high value of IDD (22.7) (Table 1). This means that commercial and services activities are very dispersed on this axis until reaching the new conurbation. It can be said that on the road parallel to the seaside between Nabeul and the city of Hammamet, the logic of urban growth can be accentuated by this proximity.

**Table 1.** Directional Dispersion Index for the four axes.

|            | IDD   | AggIDD% |
|------------|-------|---------|
| North Axis | 8.27  | 17%     |
| South Axis | 9.21  | 19%     |
| East Axis  | 8.46  | 17%     |
| West Axis  | 22.70 | 47%     |

Thus, for the interval [0, 400], it should be noted that the Directional Dispersion Indexes values are low, which means that these values are not very spread around the mean center (with an equal distribution on the four axes). Indeed, there is a high concentration of activities in the central area. This concentration decreases at the periphery, where the dispersion index values rise. Beyond the 400 m threshold, we observe a very high value of the dispersion index on the western axis, while the eastern axis represents the lowest dispersion value. This could lead us to conclude that the spatial system is becoming increasingly fragmented, particularly along the western axis (see Table 2 and Figure 9).

**Table 2.** Directional Dispersion Index measurements along the axes.

|            | IDD < 400 | AggIDD% | IDD > 400 | AggIDD% |
|------------|-----------|---------|-----------|---------|
| North Axis | 5.49      | 26%     | 8.59      | 23%     |
| South Axis | 5.24      | 24%     | 5.20      | 14%     |
| East Axis  | 5.03      | 25%     | 1.22      | 3%      |
| West Axis  | 5.60      | 26%     | 22.2      | 60%     |

### 3.3. Accessibility and Integration

By observing the global integration maps of the city of Nabeul (Figure 10), we note the centrality of the linear type of the urban network that was already apparent in 1975. This linear pattern is represented by the main axis which has a high integration value ($R_n$ = 3.13). This observation is supported by a large concentration of activities in the central area. Another integrated axis is the south one. It runs perpendicular to the main axis and leads

to the coast. Indeed, the axial map reveals an asynchronous degree of integration between the center and the periphery, where the majority of axes are segregated. On the other hand, in the 2018 map, the integration measure shows a similar integrated road system around the value of $R_n$ = 2.30 on the four axes and inside a core area of 50 ha. Beyond that, the northern axis leading to Tunis, the southern leading to the shore and the western axis leading to Hammamet have the highest integration values since they have the fewest total number of direction changes in comparison to all other streets ($R_n$: 2.15, 2.21, 2.30). We can observe that a new grid pattern is forming. This pattern is supported by the measure of integration, which represents the "to-movement" potential, and the result is predictable, considering the attractiveness of the coast in the south and the service connections that the city of Nabeul has with the two cities of Tunis (in the north) and Hammamet (in the west).

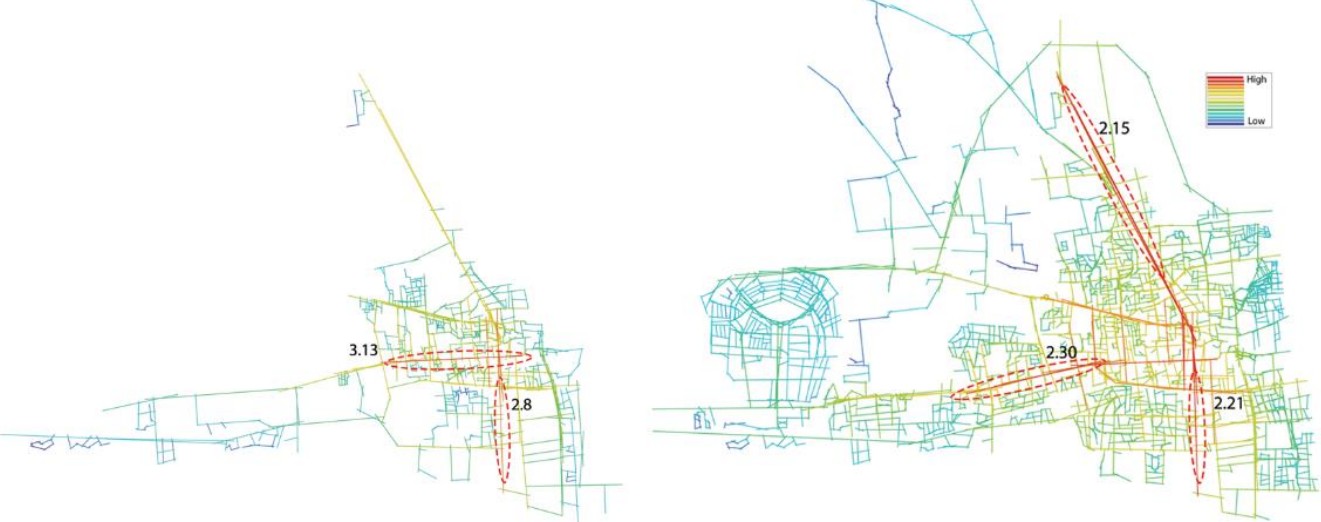

**Figure 10.** Integration ($R_n$) maps in 1975 and 2018.

The interest of the local scale analysis is that it can reveal the relationship between each axis and its near surroundings. According to the 1975 map (Figure 11), it appears that the main street of the city has the highest local integration value ($R_3$ = 4.42), while South Avenue denotes a significant connection between the center of the city and the shore. Another interesting observation is that the integration value in the west axis decreases as we move away from the central area [400, +∞]. It also appears that the peripheral roads of the city limits are the most segregated ones. In contrast, the axial map of 2018 reveals a persistent center of integration with high value in the west axis ($R_3$ = 4.32). During this phase, the network's spatial expansion was accompanied by an increase in the number of axial lines, resulting in a substantial number of streets with higher values and stronger connections to the main axis of the system. A new line segment ($R_3$ = 4.54) has appeared in the west axis, which aids in maintaining connectivity between the center and the new residential area on the west side of the city. The movement of people is probably supported by this new part of the city, which could explain the presence of so many activities scattered throughout the western axis.

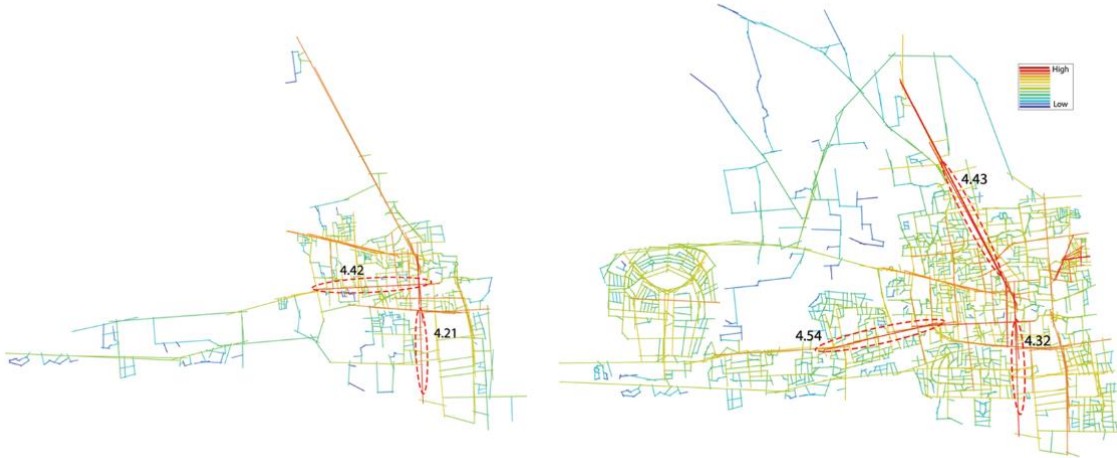

**Figure 11.** Integration ($R_3$) map in 1975 and 2018.

### 3.4. The Emergence of a Linear and Diffuse Spatial Configuration

The choice maps shown in Figure 12 show that the western street in 1975 had a high choice value Rn = 128,930. This segment is located in the central area next to the Old Medina, where there is a concentration of commercial activities and mixed uses. This axial line has the most passing trade potential and was often used like a pedestrian road. The 2018 axial line map analysis reveals a new segment along the western axis, which appears to have the highest choice measure Rn = 770,015 in 2018. This road is in fact the most important connection between the old urban fabric (the medina) and the extension of the city. Moreover, this road plays a major role in supporting the circulation flows of people in the city. This axis also shapes the distribution of activities in a very contrasting linear dispersion at the scale of the city. In fact, this linear growth seems to anticipate the emergence of new implementations along this axis.

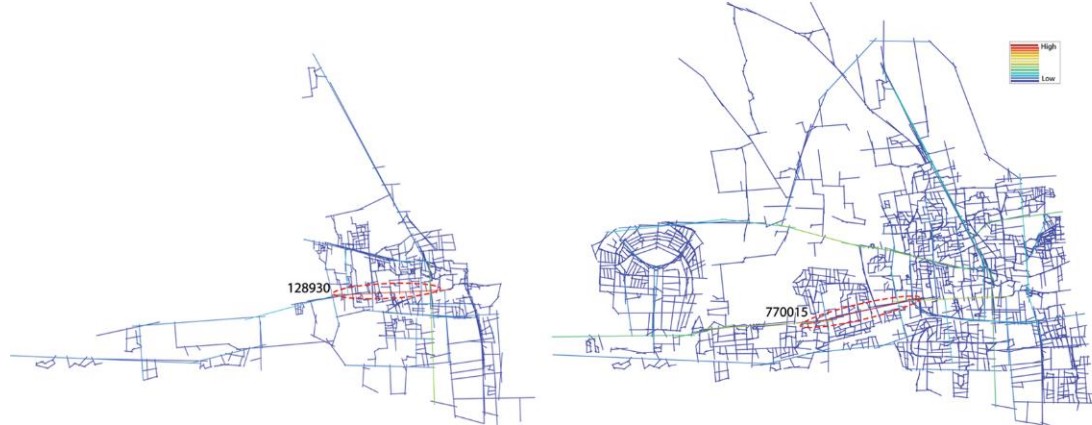

**Figure 12.** Choice (HH) map in 1975 and 2018.

This observation is corroborated by local integration measures, which show that the flow dynamics are accentuated with the appearance of a segment in the interval [400, +∞], supporting the hypothesis of a sprawl from the center towards the new seaside district installed at the western edge of the city. We denote the emergence of fragmentation on the urban fabric beyond the central area.

Specifically, one aspect that emerged in the visual confrontation can be considered through the syntactic approach. The west axis is represented by two integration values ($R_n$ = 2.19 in the central area and $R_n$ = 2.30), as well as two choice values (37,140/770,015) (Table 3). These numbers indicate statistically significant variations in the dispersion index (IDD 5.60 and 22.2). The values found in the second segment of the western axis are

certainly higher than those of the segments of the central square. These values show that this tourism axis is changing the road network's structure, as a result of how attractive this direction has become.

**Table 3.** Syntactic measurements of the four axes in 2018.

|  | Integration $R_n$ | Integration $R_3$ | Choice HH |
|---|---|---|---|
| North Axis | 2.148 | 4.436 | 54,400 |
| South Axis | 2.216 | 4.326 | 86,540 |
| East Axis | 1.950 | 4.294 | 37,140 |
| West Axis | 2.193/2.308 | 4.326/4.544 | 37,140/770,015 |

It appears that these two approaches explore the spatial organization from different points of view and give a better understanding of the situation. As a result, the probability of a double movement of concentration and dispersion can be detected.

*3.5. Different Logics of Space Production*

The spatial structure of the city of Nabeul is being altered by coastalization processes that affect all areas of urban life: social, cultural, economic and urban planning. We can say that the hypothesis of non-homogeneous dispersion is therefore retained.

Based on the different axial or activity dispersion maps generated and the underlying activity locations that underpin these patterns, the integration core is gradually being confirmed in the city of Nabeul (Figure 10). The results show that this centrality is of linear type in 1975, corresponding to R1 in the proposed model in Figure 13. Therefore, the dispersion of activities generated a concentration of activities along the primary axes. However, the concentration of activities along this main axis also suggested the parallel appearance of peripheral generated [c1] axes, corresponding to $R_2$ in Figure 13, with a progressive concentration of activities along this axis and the formation of emerging polarities. This new centrality pattern reveals a network centrality, giving rise to an integration core in the whole city. Global integration analysis indicates these emerging space-syntactic cores of a spatial network, which correlate to the city's main life center. This new active core implies the aggregation of those functions such as small and medium-sized stores and catering, which on average occupy highly efficient spaces at the global and local scales. The spatial configuration also shapes their spatial distributions.

The correlation between the dispersion index and the syntactic results is confirmed; the layout of the buildings and how the street network is organized impact social relationships and economic activities. The city's physical structure is closely connected to these social and economic interactions, and space appears as an active network of settlement processes. Consequently, one can identify a gradual transition towards a central core, where proximity to the center is constantly sought (i.e., by agglomeration forces). Thus, the center seems to maintain its symbolic functions in contrast to the "classical" model of urban landscape evolution. However, new dynamics occur as a result of the significant axes that expand the structure (e.g., $R_2$ in Figure 10). The emergence of polarities can be observed on the western axis, which has been propelled by the development of a coastal conurbation in the region of Nabeul-Hammamet and consolidated by the Mrazga project created ex Nihilo. Finally, one can conclude that the analysis of the distribution of activities complements the conclusion of the syntactic analysis of the urban space. The aim is to define configurational parameters that can be used as a prospective tool for strategic planning, as well as to define urban development trajectories for better-integrated management of urban areas.

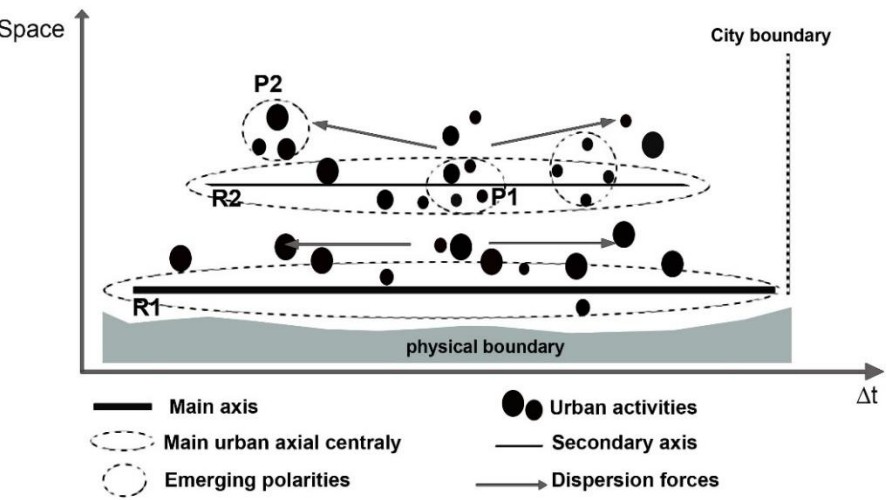

**Figure 13.** The model of urban evolution from the axial centrality to the core centrality.

## 4. Discussion

Over the past few years, space syntax analysis has provided an effective experimental way to understand spatial configuration [52]. The main advantage of a syntactic modeling approach is that it provides a practical way of offering a simplified and legible representation of the complexity of the structure of an urban system [55]. A key space syntax finding is that the function of the city is significantly correlated with the structure of space and natural movement [56,57]. However, other studies have shown that other morphological and metric properties should be also considered [33–58]. It has been demonstrated that socio-economic activities and physical configuration, as denoted by space syntax, topological and metrics, play a significant role in the evolution of a Mediterranean city [59,60]. Therefore, our study introduces a directional index of dispersion, whose objective is to take into account the location, distribution and orientation of activities on the road network. The objective is to evaluate the distribution of socio-economic activities in the urban layout and then to correlate them with structural properties revealed by space syntax measures.

In addition, an urban sprawl model has been designed that takes into account the dynamics of road network expansion [61]. The results show that the city of Nabeul is distinct in both its spatial and temporal dimensions, with its expansion guided by a logic of Littoralization that leads to the formation of a city composed of homogeneous fragments juxtaposed. At first glance, Nabeul encompasses a concentric urban model including all urban functions: not only housing but also services, public facilities and tourism activity. Therefore, these elements are dispersed but highly connected to each other: the city becomes diffuse and develops along a pre-existing territorial grid. This model differs from the common forms of evolution of coastal cities [62]. Indeed, we are witnessing the emergence of a new model of coastal city evolution, in which landscape obstacles on the one hand (i.e., coastline and mountains) and development policies on the other have contributed to an axial spatial diffusion leading to the merger of two neighboring cities (Nabeul and Hammamet) whose suburbs eventually join to form a conurbation. In contrast, the city of Monastir, taken in a previous study [18] as a representative example of a Sahelian city, has experienced concentric growth by adding new surfaces to the central urban core as a result of urban planning interventions and directed construction but also of spontaneous and self-emerging development in a second phase. This has contributed to the appearance of new structural elements such as an "emerging centrality core".

The settlement structure was first built in networks, then in "islands", with connections between them still being sparse in the second analysis period. The facilities in these earliest nuclei, which were mostly residential in nature, were limited to local services and daily shopping, with little in the way of communal public services. Only later, as the needs

of this metropolitan zone grow, do the communication infrastructure and public facility offerings become more complicated. This period corresponds to the city's most visible and spectacular metamorphosis, in which the spatial structure is understood as changes in the configuration of the built space. The space syntax implemented in this study, which was complemented by a novel measure of directional dispersion, shows its fragmented forms: the appearance of these "islands" in the peri-urban space that essentially constitute the diffuse spot. Both the computational metrics and structural model reveal some intriguing patterns. A nonlinear degree-strength connection means that the accelerating growth of these weights deserves more consideration in the evolution of the urban networks [1]. This study found significant correlations between the dispersion index and the metrics of the syntactic analysis, such as street circularity and choice measures, which help us better understand how these connections shape the structure and affect the performance of a weighted evolving network. In the development of these ideas, dispersion presents the fundamental component of the phenomenon of urban sprawl in the city of Nabeul. This dispersion is responsible for a large part of the spatial structure. This empirically confirms that coastal cities exhibit atypical sprawl patterns.

The findings also highlight that different urban dispersion, or functional centrality and the emergence of "hybrid" centrality [18], are the outcomes of differentially structured developments and planning policies. In fact, unlike Monastir, the city of Nabeul develops a different model of urban evolution. The spatial structure of Nabeul has metamorphosed according to regions parallel to the coastline forming an integration nucleus with varying radii. This work could be extended to similar coastal environments, by still combining spatial syntactic techniques with functional and statistical measures, where the predictive potential of both approaches provides a potential support for urban planning and studies.

## 5. Conclusions

Finally, this experimentation addresses the broader topic of the manufacturing of new urban polarities, and it suggests perspectives for integrated management of coastal cities. We believe that this modeling approach can favor a better understanding of the complex relationships between land use and structural properties of a coastal city. The correlation between dispersion indices and space syntax figures provides a new way of characterizing the distribution of human activities and land development. This modeling approach, as applied to the city of Nabeul, presents a novel technique for empirically evaluating the complexity of structural patterns as they appear and evolve in the city and provides a specific reference framework for coastal cities in Tunisia and beyond.

The proposed urban dispersion model still needs further development. In particular, the underlying modeling principles should be extended to the temporal dimension. As it stands, the Directional Dispersion Index is derived for a specific linear axis over a given time. Therefore, while dealing with multiple temporalities and even scales, there is a need to apply the dispersion index to the evolution of the whole urban space at different levels of granularity. Driving factors at different spatial levels and across time might then be revealed.

Overall, this study also showed that configurational and physical barriers (i.e., Mediterranean Sea and the rivers) play an important role in the city of Nabeul's growth and distribution of activities. Clearly the city of Nabeul has undergone critical transformation phases as evidenced by the evolution of its structural layout and socio-economic variables. The study's findings may provide valuable insights to urban planners and authorities by improving their understanding of the city's evolution patterns and threats in relation to physical and economic variables. This modeling approach can be generalized toward an analytical framework to capture empirical realities as a component of strategic visions in territorial development policies.

Finally, from a methodological standpoint, this research combines a theoretical and computational geomatics approach applied to the field of urban studies. It shows the potential of a sound integration of geographical and semantic data using a modelling and structural representation derived and extended from space syntax principles. It also favors

the analysis and comprehension of emerging urban properties and patterns at different scales in space and time.

**Author Contributions:** N.B.: Conceptualization, methodology, software, validation, formal analysis, investigation, writing—original draft preparation. A.R.B.: Methodology, software, validation, formal analysis, investigation, writing—original draft preparation. C.C.: Formal analysis, writing—review and editing, visualization, supervision, resources, data curation. H.R.: Writing—review and editing, visualization, supervision, resources, data curation. All authors have read and agreed to the published version of the manuscript.

**Funding:** This research received no external funding.

**Institutional Review Board Statement:** Not applicable.

**Informed Consent Statement:** Not applicable.

**Data Availability Statement:** The acquired data will be used exclusively for this research.

**Conflicts of Interest:** The authors declare no conflict of interest.

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
