# Peer review of "A Dispersion Index for the Analysis of the Distribution of Activities in the Tunisian Coastal City of Nabeul"

_2673-7418, doi:10.3390/geomatics2020010_

Round 1
Reviewer 1 Report
A dispersion index for the analysis of the distribution of activities in the Tunisian coastal city of Nabeul
The authors should address the following comments to improve the quality of the paper:
• Please enlarge and improve the quality of all figures to become more legible.
• Lines 15-21: The study area is previously mentioned as “the city of Nabeul.” Please avoid repeating the same phrase in lines 15 and 21. Instead, write “the city.”
• Line 23: The abstract should conclude with a sentence on the implications of the finding on planning and managing the city.
• Lines 31-35: Please expand more on the factors that attract growth to coastal areas.
• Lines 63-65: There is the need to review a theory of urban spatial structure to explain the forces behind the pattern of growth, activities, and configuration of the study area
• Line 117: It is better to present the population living in communal areas in percentage instead of absolute numbers.
• Section 2: Please justify all methodological choices, including selected models and study duration (1975-2018), equations 1-7, analytical techniques, and provide full references to data sources.
• Section 4: Please improve the discussion by highlighting how the study findings support or differ from similar prior studies and likely explanations. Only two prior studies [1, 16] are currently cited in the section
• The discussion section should also link the study findings to the research questions, underscore the value the study adds to the literature, and indicate whether the modeling approach can be generalizable/applied in other settings.
• Please expand the conclusion section to state the implications of the study on urban planning/management, policy, and practice.
Author Response
Reviewer #1 Comment #1: Please enlarge and improve the quality of all figures to become more legible.
Response: Thank you very much for your suggestion, the quality of all figures has been improved and their size enlarged as suggested.
Reviewer #1 Comment #2: Lines 15-21: The study area is previously mentioned as “the city of Nabeul.” Please avoid repeating the same phrase in lines 15 and 21. Instead, write “the city.”.
Response: Thank you very much for this suggestion, corrected.
Reviewer #1 Comment 3: Line 23: The abstract should conclude with a sentence on the implications of the finding on planning and managing the city.
Response: Thank you for this valuable suggestion. The final sentence of the abstract has been rewritten as follows: “Finally, this study introduces a novel approach for identifying urban structural polarity and activities, as well as new perspectives for coastal land management and planning.”
Reviewer #1 Comment 4: Lines 31-35: Please expand more on the factors that attract growth to coastal areas.
Response: Thank you for this valuable suggestion. This suggestion has been considered and specific writings have been inserted to clarify this. Lines 33 to 35 are as follows: “Mediterranean coastal locations are very appealing due to their year-round pleasant weather [3]. Therefore, the fast-growing development of seaside tourism has largely contributed to the image of the coastline and its urban dynamism [4].”
Reviewer #1 Comment 5: Lines 63-65: There is a need to review a theory of urban spatial structure to explain the forces behind the pattern of growth, activities, and configuration of the study area.
Response: Thank you for this suggestion. An additional review of background theory and the relationship between urban structures, driving forces and activities have been inserted in lines 65 to 74 as follows: “A thorough modelling of urban growth processes is a prerequisite to a sound understanding of urban dynamics, as well as better management and planning tasks [19-20]. Several models have been developed to describe urban structural properties, including the concentric zone [21], sector-based [22], multi-nuclei [13] and central place approaches [24]. Recent urban sprawl studies simulated land-use changes using cellular automata, complex systems theory, and agent-based systems and characterizes them at a fine spatio-temporal level [25-26]. However, most of these works concentrate on urban structural properties [1-27], rather than socio-economic patterns and human behaviors [19]. Therefore, several lines of research have explored a closer connection between spatial configuration, land use, and urban economics [28].”
Reviewer #1 Comment 6: Line 117: It is better to present the population living in communal areas in percentage instead of absolute numbers.
Response: Thank you for this wise suggestion. This has been considered and modified accordingly.
Reviewer #1 Comment 7: Section 2: Please justify all methodological choices, including selected models and study duration (1975-2018), equations 1-7, analytical techniques, and provide full references to data sources.
Response: Thank you for these suggestions. The methodological choices and motivation have been further discussed in the paper as follows: Lines 162 to 169: “The years from 1975 to 2018 can help us understand the historic development process and thus capture the spatial structure’s evolution.1975 refers to a period of urbanization marked by tourist and industrial expansion, as a result of government economic policies conducted in the 1970s [7-44-45]. The year 2018 is then chosen as a representative recent date on which Nabeul has experienced extremely violent storms and torrential rainfall that have challenged the urban planning of the city. All geographical data is based on aerial photographs from 1975, as well as prior research [7] and recent Google Earth images.’ Lines 205 to 209 “Finally, combining the statistical index and structural analysis within a single approach, as depicted in Figure 4, allows for an integration of complementary geographic and socio-economic data to be confronted to spatial syntax figures [49]. This enables us to investigate the relationship between commercial activity distribution and the emergence of new forms of spatial organization caused by coastal attractiveness.’ Additional references have been given to equations when appropriate: Lines 225 to 226: “For instance, in (1), ID(j) is given by the mean and standard deviation of the activity’s distribution.” Line 309: (6) “Relative Asymmetry has been introduced by Hillier and Hanson [27].” Lines 322 to 323: (7) by Hillier et al.in a paper called ‘Creating life: or does architecture determine anything?’ [53]
Reviewer #1 Comment 8: Section 4: Please improve the discussion by highlighting how the study findings support or differ from similar prior studies and likely explanations. Only two prior studies [1, 16] are currently cited in the section
Response: Thank you for this wise suggestion. Methodological differences with previous studies have been further discussed along lines 472 to 485. “Over the past few years, space syntax analysis has provided an effective experimental way to understand spatial configuration [55]. A key space syntax finding is that the function of the city is significantly correlated with the structure of space and natural movement [57-58]. However, other studies have shown that other morphological and metric properties should be also considered [59-33]. It has been demonstrated that socio-economic activities and physical configuration, as denoted by space syntax, topological and metrics, play a significant role in the evolution of a Mediterranean city [60-61]. Therefore, our study introduces a directional index of dispersion, whose objective is to take into account the location, distribution and orientation of activities on the road network. The objective is to evaluate the distribution of socio-economic activities in the urban layout and then correlate it with structural properties revealed by space syntax measures.
Reviewer #1 Comment 9: The discussion section should also link the study findings to the research questions, underscore the value the study adds to the literature, and indicate whether the modelling approach can be generalizable/applied in other settings.
Response: Thank you for this wise suggestion. The final discussion has been partly rewritten to highlight the value of our research as related to previous work, research questions identified in the introduction and how the whole approach might be generalized to other urban contexts lines 529 to 531: “This work could be extended to similar coastal environments, by still combining spatial syntactic techniques with functional and statistical measures, where the predictive potential of both approaches provides a potential support for urban planning and studies.”
Reviewer #1 Comment 10: Please expand the conclusion section to state the implications of the study on urban planning/management, policy, and practice.
Response: Thank you for this wise suggestion. The conclusion now discussed the possible impact and interest of our work for urban planning and management as well as for decision-makers, cf. lines 549 To 557 as follows: “Overall, this study showed that configurational and physical barriers (i.e., Mediterranean Sea and the rivers) play an important role in the city of Nabeul’s growth and distribution of activities. Clearly, the city of Nabeul’s evolution has undergone critical transformation phases as evidenced by the evolution of its structural layout and socio-economic variables. The study's findings may provide valuable insights to urban planners and authorities by improving their understanding of the city's evolution patterns and threats in relation to physical and economic variables. This modelling approach can be generalized toward an analytical framework to capture empirical realities as a component of strategic visions in territorial development policies”

Reviewer 2 Report
Your paper could improve in a number of areas such as a more thorough discussion of the design, development, testing and evaluation results; clarifying the key significance of the research contribution; ascertaining that the research fits the aims and scope of the journal; and a better command and flow of English writing throughout the paper.
The references should be updated with the most recent in your paper's research field of relevance. I recommend the authors to consult the following survey and empirical papers to contextualize your findings. This should help the readers to understand the novelty of your work.
Improving Malicious URLs Detection via Feature Engineering: Linear and Nonlinear Space Transformation Methods, Information Systems (2020), DOI: https://doi.org/10.1016/j.is.2020.101494
A bibliometric review of cryptocurrencies: how have they grown?. Financ Innov 8, 2 (2022). https://doi.org/10.1186/s40854-021-00306-5
Author Response
Reviewer #2 General Comment #1: Your paper could improve in a number of areas such as a more thorough discussion of the design, development, testing and evaluation results; clarifying the key significance of the research contribution; ascertaining that the research fits the aims and scope of the journal; and a better command and flow of English writing throughout the paper.
Response: Thank you very much for your valuable suggestions. First of all, careful cross-checking of the language has been made. Next, we made every effort to clarify the findings as well as their evaluation. Finally, the research contribution and its interest in the scope of the journal have been clarified in the final discussion and conclusion. The contribution of the paper has been clarified in several places (cf. file with responses to all reviewers): • abstract: cf response to reviewer 1 first comment • relation to previous work: cf. response to reviewer 1 comment 5 • methodological choices: cf. response to reviewer 1 comment 7 • discussion on the findings: cf. responses to reviewer 1 comments 8 to 10 A final paragraph has been inserted in the conclusion to highlight the contribution to the field of geomatics as follows: “Finally, from a methodological standpoint, this research combines a theoretical and computational geomatics approach applied to the field of urban studies. It shows the potential of a sound integration of geographical and semantic data using a modelling and structural representation derived and extended from space syntax principles. It also favors the analysis and comprehension of emerging urban properties and patterns at different scales in space and time.”
Reviewer #2 General Comment #2: the references should be updated with the most recent in your paper's research field of relevance.
Response: several additional references have been inserted in the paper: -Pasquali, D.; Marucci, A. The Effects of Urban and Economic Development on Coastal Zone Management. Sustainability 2021, 13, 6071. https://doi.org/10.3390/su13116071 -Liziard, S. Littoralisation de l’Arc Latin : Analyse Spatio-Temporelle de La Répartition de La Population à Une Échelle Fine. Espace populations sociétés 2013, No. 2013/1-2, 21–40. https://doi.org/10.4000/eps.5308. -Cheng,J.; Masser,I. Modelling Urban Growth Patterns: A Multiscale Perspective. Environment and Planning A: Economy and Space 2003, 35 (4), 679–704. https://doi.org/10.1068/a35118. -Burgess, E. W. The Growth of the City: An Introduction to a Research Project. InThe City. 1923, 47–62. -Adams, J. S. Hoyt, H. 1939: The Structure and Growth of Residential Neighborhoods in American Cities. Washington, DC: Federal Housing Administration. Progress in Human Geography 2005, 29 (3), 321–325. https://doi.org/10.1191/0309132505ph552xx. -Harris, C. D.; Ullman, E. L. The Nature of Cities. The ANNALS of the American Academy of Political and Social Science 1945, 242 (1), 7–17. https://doi.org/10.1177/000271624524200103. -Getis, A.; Getis, J. Christaller’s Central Place Theory. Journal of Geography 1966, 65 (5), 220–226. https://doi.org/10.1080/00221346608982415. -Mozaffaree Pour, N.; Oja, T. Urban Expansion Simulated by Integrated Cellular Automata and Agent-Based Models; an Example of Tallinn, Estonia. Urban Science 2021, 5 (4), 85. https://doi.org/10.3390/urbansci5040085. -Portugali, J. Self-Organization and the City; Springer Berlin Heidelberg: Berlin, Heidelberg, 2000. https://doi.org/10.1007/978-3-662-04099-7 -Narvaez, L.; Penn, A.; Griffiths, S. The spatial dimensions of trade: From the geography of uses to the architecture of local economies. ITU J. Fac. Archit. 2014, 11, 209–230. -Hellal, M. L’évolution Du Système Touristique En Tunisie. Perspectives de Gouvernance En Contexte de Crise. Études caribéennes 2020, No. 6. https://doi.org/10.4000/etudescaribeennes.19397. -Dhaher, N. L’aménagement Du Territoire Tunisien : 50 Ans de Politiques à l’Épreuve de La Mondialisation. EchoGéo 2010, No. 13. https://doi.org/10.4000/echogeo.12055. -Sanders, L. Modèles en Analyse Spatiale: introduction 2001, 17–29 -Hillier, B.; Burdett, R.; Peponis, J.; Penn, A. Creating Life: Or, Does Architecture Determine Anything? Architecture & Comportement/Architecture & Behaviour 1986, 3 (3), 233–250 -Jiang, B.; Claramunt, C.; Klarqvist, B. Integration of Space Syntax into GIS for Modelling Urban Spaces. International Journal of Applied Earth Observation and Geoinformation 2000, 2 (3-4), 161–171. https://doi.org/10.1016/s0303-2434(00)85010-2. -Dettlaff, W. Space syntax analysis – methodology of understanding the space PhD Int. J., 1 2014, pp. 283-291. http://sdpg.pg.gda.pl/pij/wp-content/blogs.dir/133/files/2014/12/01_2014_30-dettlaff.pdf (accessed 2022 -01 -06) -Hillier, B.; Penn, A.; Hanson, J.; Grajewski, T.; Xu, J. Natural Movement: Or, Configuration and Attraction in Urban Pedestrian Movement. Environment and Planning B: Planning and Design 1993, 20 (1), 29–66. https://doi.org/10.1068/b200029. -Karimi, K. A Configurational Approach to Analytical Urban Design: “Space Syntax” Methodology. URBAN DESIGN International 2012, 17 (4), 297–318. https://doi.org/10.1057/udi.2012.19. -Ye, Yu, and Akkelies Van Nes. "Making spatial diagnosis in combining Space Syntax, Spacematrix and MXI with GIS of new and old towns." EAAE/ISUF International Conference 2012. 2012. -Atakara, C.; Allahmoradi, M. Investigating the Urban Spatial Growth by Using Space Syntax and GIS—a Case Study of Famagusta City. ISPRS International Journal of Geo-Information 2021, 10 (10), 638. https://doi.org/10.3390/ijgi10100638. -Shpuza, E. Allometry in the Syntax of Street Networks: Evolution of Adriatic and Ionian Coastal Cities 1800–2010. Environment and Planning B: Planning and Design 2014, 41 (3), 450–471. https://doi.org/10.1068/b39109. -Batty, M.; Xie, Y.; Sun, Z. Modeling Urban Dynamics through GIS-Based Cellular Automata. Computers, Environment and Urban Systems 1999, 23 (3), 205–233. https://doi.org/10.1016/s0198-9715(99)00015-0. -Barragán, J. M.; de Andrés, M. Analysis and Trends of the World’s Coastal Cities and Agglomerations. Ocean & Coastal Management 2015, 114, 11–20. https://doi.org/10.1016/j.ocecoaman.2015.06.004.
Reviewer #2 General Comment #2: I recommend the authors to consult the following survey and empirical papers to contextualize your findings. This should help the readers to understand the novelty of your work. Improving Malicious URLs Detection via Feature Engineering: Linear and Nonlinear Space Transformation Methods, Information Systems (2020), DOI: https://doi.org/10.1016/j.is.2020.101494 A bibliometric review of cryptocurrencies: how have they grown?. Financ Innov 8, 2 (2022). https://doi.org/10.1186/s40854-021-00306-5.
Response: Thanks for these suggestions that have been useful to us when providing more support to the presentation of the findings.
